**Data Availability Statement:** All relevant data are within the paper.

**Funding:** The authors received no specific funding for this work.

# Identification of key genes and immune infiltration modulated by CPAP in obstructive sleep apnea by integrated bioinformatics analysis

**Cheng Fan, Shiyuan Huang, Chunhua Xiang, Tianhui An, Yi Song ⓘ** *

Department of Geriatrics, Union Hospital, Tongji Medical College, Huazhong University of Science and Technology, Wuhan, China

* yisong@hust.edu.cn

## Abstract

Patients with obstructive sleep apnea (OSA) experience partial or complete upper airway collapses during sleep resulting in nocturnal hypoxia-normoxia cycling, and continuous positive airway pressure (CPAP) is the golden treatment for OSA. Nevertheless, the exact mechanisms of action, especially the transcriptome effect of CPAP on OSA patients, remain elusive. The goal of this study was to evaluate the longitudinal alterations in peripheral blood mononuclear cells transcriptome profiles of OSA patients in order to identify the hub gene and immune response. GSE133601 was downloaded from Gene Expression Omnibus (GEO). We identified black module via weighted gene co-expression network analysis (WGCNA), the genes in which were correlated significantly with the clinical trait of CPAP treatment. Finally, eleven hub genes (TRAV10, SNORA36A, RPL10, OBP2B, IGLV1-40, H2BC8, ESAM, DNASE1L3, CD22, ANK3, ACP3) were traced and used to construct a random forest model to predict therapeutic efficacy of CPAP in OSA with a good performance with AUC of 0.92. We further studied the immune cells infiltration in OSA patients with CIBERSORT, and monocytes were found to be related with the remission of OSA and partially correlated with the hub genes identified. In conclusion, these key genes and immune infiltration may be of great importance in the remission of OSA and related research of these genes may provide a new therapeutic target for OSA in the future.

## Background

OSA is a highly prevalent sleep disorder, characterized by repetitive partial or complete airway collapse resulting in apneic events during sleep, sleep fragmentation and chronic intermittent hypoxia. The direct consequence of the intermittent hypoxia is an oxidative imbalance, with reactive oxygen species production, inflammatory cytokines (IL2, IL4, IL6), lipid peroxidation and cell-free DNA production [1], and patients with OSA have higher risk of developing metabolic comorbidities and cardiovascular disorders [2–4]. CPAP, as the first-choice common

**Competing interests:** The authors have declared that no competing interests exist.

treatment for OSA, may exert various additional therapeutic benefits beyond its improvement of the oxygen supply. More specifically, CPAP treatment significantly changed the profile of endothelial function, improved blood flow in both macro- and microcirculation and lowered the inflammatory mediator level in a review conducted by Klaudia et al [5]. In addition, proper use of CPAP therapy provided significant benefits for the treatment of arrhythmia in patients with OSA [6]. However, CPAP is not to date the only possible treatment in patients with obstructive sleep apnea. CPAP treatment could fail in patients with lingual tonsil hypertrophy and transoral robotic surgery (TORS) could represent a valid therapeutic option for retrolingual airway collapse [7].

The etiology of OSA is not well established fundamentally, although several possible molecular mechanisms have been proposed. For instance, it was reported that early-morning after one night of intermittent hypoxemia-induced augmentation of DUSP1 gene expression could attenuated by CPAP treatment [8]. With the rapid development of microarray [9] and high-throughput gene sequencing technologies [10], gene expression-based disease biomarkers have attracted more and more attention. Previous studies had identified subcutaneous adipose tissue transcriptional pattern modulated in OSA and in response to its effective treatment of CPAP [11]. Nevertheless, reports about transcriptome in peripheral blood mononuclear cells affected by CPAP are limited. Because polysomnography (PSG) is expensive and time-consuming in clinical application, it is necessary to find cheap and convenient biomarker from venous blood for the therapeutic efficacy monitoring of CPAP intervention.

To identify the genes regulated by CPAP, we downloaded relevant dataset (GSE133601) from GEO and constructed weighted co-expression network by WGCNA method [12]. Key modules closely associated with CPAP therapy were identified. Functional enrichment analysis was further used to hint the biological function of the key module genes. Next, we identified hub genes by searching for common genes of differentially expressed genes (DEGs) and black module genes. Since CIBERSORT is a widely used analysis tool using microarray data or RNA-seq data to investigate the expression profile of 22 types of immune cells and to calculate the proportions of each type of immune cells in the samples [13], immune cells infiltration was evaluated in OSA patients via CIBERSORT. Furthermore, correlations between key genes and immune cells infiltration were calculated. To the best of our knowledge, this is the first study that integrated bioinformatics analyses are used to find key genes and immune infiltration characteristics among peripheral blood mononuclear cells modulated by CPAP in OSA patients.

## Materials and methods

### Data download and preprocessing

First, we researched with ("continuous positive airway pressure" or "CPAP") and ("whole blood" or "peripheral blood") and ("obstructive sleep apnea" or "OSA") in GEO database to find the interested expression data. After excluding irrelevant datasets, the dataset of GSE133601 was included. In this gene expression study, 15 individuals of obstructive sleep apnea were treated with CPAP therapy (defined by at least 4 hours of CPAP use over the 3-month intervention period) participated. The GSE133601 contained 30 samples of peripheral blood mononuclear cells, which including pre and post treatment of CPAP. After the microarray data were downloaded successfully, it was transformed into gene expression information by the expression value of probes from the GEO dataset. Probes with more than one gene were eliminated and the average value was calculated for genes corresponding to more than one probe. Data analysis procedures for the whole study were illustrated in Fig 1.

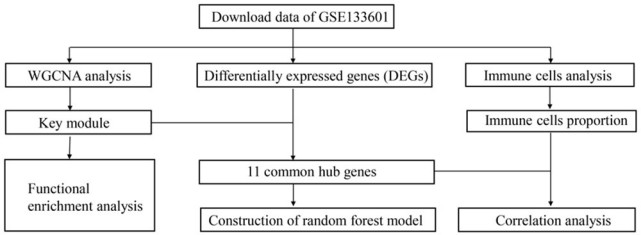

**Fig 1. Flow chart for the whole study.**

## Weighted gene co-expression network analysis (WGCNA)

Co-expression modules were constructed using 5,000 genes of high expression values by the "WGCNA" package [12] in R. Soft-thresholding parameter of β = 12 was selected to ensure a scale-free network, when the degree of independence was 0.8. The corresponding gene information for each module was extracted and the minimum number of genes was set as 50 to achieve high reliability of the results. The interaction of co-expression modules and module-trait relationship based on the eigengene were determined using R language [14] and the WGCNA algorithm. The heatmap tools package was used to analyze the strength of the interactions. The module with the first-ranked module significance (MS) was considered as the clinically significant modules.

## Functional enrichment analysis

To investigate the function of the genes in clinically significant modules, the Gene Ontology (GO) annotation [15] and Kyoto Encyclopedia of Genes and Genomes (KEGG) pathway enrichment analyses [16] were performed by DAVID database (https://david.ncifcrf.gov/). The GO functional analysis was divided into the following three parts: biological process (BP), molecular function (MF) and cell component (CC).

## Identification of differentially expressed genes (DEGs) and common genes

A heat map was drawn to exhibit the expression difference between two groups of the top 200 DEGs with the "limma" package [17]. Common genes in both the black module and DEGs obtained in GSE133601 were identified using the VennDiagram package [18] in R.

## Random forest

The random forest is a classification method that uses multiple trees to train and predict samples with high accuracy [19]. And it is a powerful tool for constructing a predictive model for new biomarkers, which was built via the randomForest package [20] in R software. It is less prone to over-fitting problems and can handle a large amount of noise. We used the common genes of the black module and DEGs as the classification feature, and pre and post CPAP samples as the variables. The samples in GSE133601 (n = 30) were randomly divided into the training set and test set with the ratio 2:1. The model built by the training set was tested by the test set. The error rate was calculated through Out-of-bag (OOB) [21] to evaluate the correct rate of the combined classification. The mean decrease accuracy (MDA) of the random forest model was positively correlated with the predictive variable, and the mean decrease Gini (MDG) was positively correlated with the most important variable [22]. Using pROC package [23], the value of area under the receiver-operator characteristics (ROC) curve (AUC) was used to validate the accuracy of the established model.

## Evaluation of immune cell infiltration and its correlation with key genes

We assessed 22 types of immune cell using the CIBERSORT algorithm [13], including macrophages, T cells, natural killer (NK) cells, mast cells, B cells, dendritic cells (DC), monocytes, plasma cells, neutrophils, and eosinophils. After obtaining the expression matrix of immune cells according to the instruction of the CIBERSORT website, we use "ggplot2" package to draw boxplot to depict the distribution of immune cells. The Pearson correlation coefficient between each kind of immune cells was calculated and the results were displayed by correlation heatmap using "corrplot" package [24] in R software.

## Results

### Construction of weighted co-expression network and identification of key modules

Weighted co-expression network was constructed by WGCNA based on the GSE133601. In summary, a total of 15 samples with post CPAP therapy and 15 corresponding paired control samples of pre-treatment were included into the analysis. We chose β = 12 as the soft-thresholding parameter (Fig 2A). Then, cluster dendrogram (Fig 2B) and network heatmap plot (Fig 2C) were displayed. Correlation between different modules was visualized among the 13 modules. Meanwhile, based on the calculation of Pearson's correlation coefficient, the black module which showed the highest MS (Fig 2D) was considered as the highest correlation one with the clinical trait of CPAP (cor = 0.31, $p$-value = 0.09) (Fig 2E). Finally, the black module, which contained 981 genes, was selected as a key module to be studied in following analysis.

### Functional enrichment analysis of genes in the black module

The black module genes are mainly enriched in exonucleolytic nuclear-transcribed mRNA catabolic process involved in deadenylation-dependent decay (ontology: BP), cytoplasm (ontology: CC), and ubiquitin-protein transferase activity (ontology: MF). The top 15 GO enrichment analysis results are shown (Table 1). Based on KEGG pathway mapping, the black module genes are involved in 7 pathways, especially in RNA degradation significantly ($P < 0.05$). As is shown in Table 2.

### Identification of common hub genes

The top 200 differentially expressed genes (DEGs) were exhibited in a heat map, including top 100 up-regulated genes and top 100 down-regulated genes (Fig 3A). A total of 11 common hub genes (TRAV10, SNORA36A, RPL10, OBP2B, IGLV1-40, H2BC8, ESAM, DNASE1L3, CD22, ANK3, ACP3) were identified for further analysis, by application of the VennDiagram package in R (Fig 3B).

### The random forest model of hub genes

The random forest regression classification model was constructed with 11 common hub genes. The false positive rate of the model fell to the lowest (about 34.49%), when mtry = 8 (Fig 4A). The error rate was almost stable, when the number of decision trees was about 500 (Fig 4B). Two indexes, mean decrease accuracy (the prediction error rate based on OOB) and mean decrease Gini (the Gini coefficient based on the sample fitting model), were calculated to measure the importance of variables. The results showed that OBP2B, IGLV1-40 and TRV10 were the more important variables (Fig 4C). We used the established model in the testing data sets to prove its high prediction cost. The diagnostic efficiency for testing set was 0.92

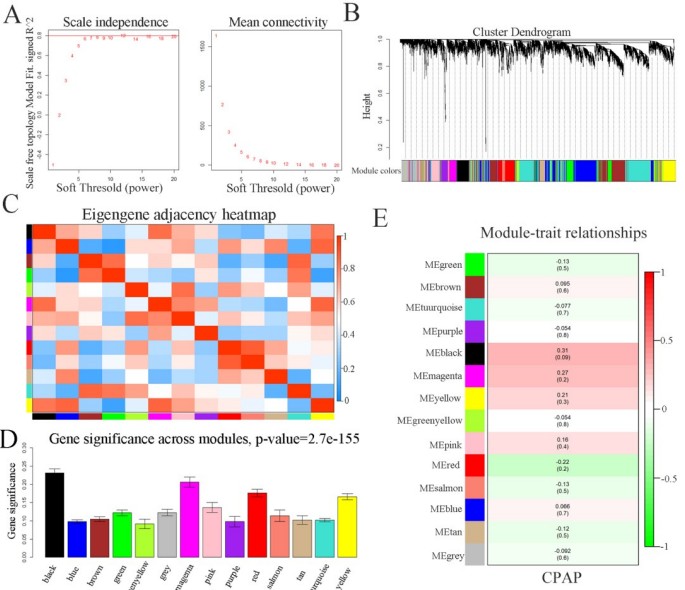

**Fig 2. Weighted correlation network analysis.** (A) Analysis of scale-free fit parameter and mean connectivity for various soft-thresholding powers. (B) Cluster dendrogram of genes, with dissimilarity based on topological overlap. (C) Co-expression similarity of entire modules based on hierarchical clustering of module eigengenes and the correlation between different modules. The color of cells in the heatmap represented the correlation coefficients of different sizes. Specifically, red colors stood for the positive correlations and blue colors represented the negative correlations. (D) Distribution of average gene significance and errors in the modules associated with the response to CPAP therapy. (E) Correlation heat map of gene modules and phenotypes. The figure without brackets in each cell indicated the clinical feature correlation coefficients. The corresponding p-value was shown below in parentheses. Red is positively correlated with the phenotype; green is negatively correlated with the phenotype.

**Table 1. Top 15 GO enrichment terms associated with genes in black module.**

| Categray | Top 15 enriched GO terms. | Count | P-value |
|---|---|---|---|
| BP | GO:0043928~exonucleolytic nuclear-transcribed mRNA catabolic process involved in deadenylation-dependent decay | 7 | 2.1E-3 |
| | GO:0007595~lactation | 8 | 3.4E-3 |
| | GO:0051384~response to glucocorticoid | 10 | 3.5E-3 |
| | GO:0045124~regulation of bone resorption | 4 | 7.3E-3 |
| | GO:0001886~endothelial cell morphogenesis | 4 | 1.3E-2 |
| CC | GO:0005737~cytoplasm | 287 | 2.4E-4 |
| | GO:0005829~cytosol | 191 | 4.7E-4 |
| | GO:0016607~nuclear speck | 19 | 5.4E-3 |
| | GO:0043204~perikaryon | 12 | 1.6E-2 |
| | GO:0042383~sarcolemma | 10 | 1.7E-2 |
| MF | GO:0004842~ubiquitin-protein transferase activity | 26 | 1.1E-2 |
| | GO:0051213~dioxygenase activity | 5 | 1.5E-2 |
| | GO:0051920~peroxiredoxin activity | 3 | 2.9E-2 |
| | GO:0005179~hormone activity | 10 | 3.0E-2 |
| | GO:0005515~protein binding | 435 | 3.1E-2 |

BP, biological process; MF, molecular function; CC, cell component.

**Table 2. Kyoto Encyclopedia of Genes and Genomes (KEGG) pathway analysis of genes in black module.**

| Pathway | ID | Gene count | P-value | Genes |
|---|---|---|---|---|
| RNA degradation | hsa03018 | 10 | 1.4E-2 | PAN2, EDC4, LSM6, CNOT7, DIS3, CNOT2, PABPC3, DCPS, PFKP, EDC3 |
| Neuroactive ligand-receptor interaction | hsa04080 | 21 | 5.9E-2 | UTS2R, LHCGR, OXTR, CALCRL, NPY5R, PTGER1, CHRNA4, MLNR, SCTR, LPAR1, TACR2, GRIK2, HTR5A, SSTR4, HCRTR2, RXFP2, P2RY8, HRH1, HRH3, LPAR6, TSHB |
| Sulfur metabolism | hsa00920 | 3 | 7.0E-2 | MPST, ETHE1, SUOX |
| Transcriptional misregulation in cancer | hsa05202 | 14 | 7.2E-2 | NGFR, CSF1R, CDKN1B, ITGAM, FUS, IL1R2, JMJD1C, MLF1, FOXO1, PTK2, SPINT1, TLX1, ITGB7, MET |
| Proteoglycans in cancer | hsa05205 | 16 | 7.2E-2 | ARHGEF12, PDPK1, SRC, LUM, ANK3, HIF1A, PTK2, ACTB, VEGFA, SMO, CTSL, TIMP3, ITGAV, PPP1R12B, MET, WNT2 |
| Ubiquitin mediated proteolysis | hsa04120 | 12 | 7.8E-2 | UBE2W, CUL5, UBA7, AIRE, FBXW11, UBE2D2, SYVN1, UBA2, ANAPC4, KEAP1, STUB1, SKP1 |
| Amino sugar and nucleotide sugar metabolism | hsa00520 | 6 | 8.8E-2 | CYB5R2, NAGK, GFPT2, GMPPA, PGM3, HK2 |

(Fig 4D). And the random forest model had a better value of AUC compared with all the single gene, in which OBP2B gene had the highest AUC of 0.813.

## Immune cell infiltration results

We used the CIBERSORT algorithm to evaluate the association between CPAP phenotype and immune cells infiltration. The relative proportion of immune cell subtypes was displayed in the stacked barplot (Fig 5A) and boxplot (Fig 5B). The results showed that monocytes had the highest proportion and the number of monocytes in post-CPAP group was higher than control group ($P < 0.05$), which suggested that monocytes were vital in OSA and the proportion of monocytes may be modulated by CPAP. However, there was no significant difference in the other 21 immune cells, let alone 5 kinds of immune cells with extremely low proportion (which were not shown). Finally, we found that CD4 naïve T cells, γδ T cells, naïve B cell, CD4 memory resting T cell and resting NK cells all had negative correlation with monocytes (Fig 5C) ($P < 0.05$). Moreover, we had calculated the correlation between monocytes and 11 hub genes. TRAV10, SNORA36A, CD22 and ANK3 had negative correlation with monocytes ($P < 0.01$), and DNASE1L3 and ACP3 had positive correlation with monocytes ($P < 0.01$).

## Discussion

CPAP has become the first line therapy for OSA for a long time. In a meta-analysis of 35 trials, CPAP showed positive results in the apnea-hypopnea index (AHI) mean difference -33.8 events in an hour [25]. Other treatments are less common. For example, oral appliances are proven to be less effective than CPAP, especially in severe OSA [26]. Since traditional efficacy monitoring method PSG is expensive and time-consuming, the establishment of predictive biomarkers for OSA therapy response has become a priority. To investigate potential biomarkers for OSA therapy, gene expression data and network-based approaches were applied. The gene co-expression network was built via the WGCNA analysis method to identify key modules related to the clinical trait (CPAP treatment). We identified 981 genes in black module, which were significantly related to the interactions of target genes induced by CPAP therapy in patients with OSA. To the end, 11 hub genes were derived by intersection of black module genes and top 200 DEGs. Furthermore, random forest prediction model was constructed. ROC curve analysis indicated that the model had a good performance compared with single gene prediction. Finally, we explored the immune infiltration characteristics of OSA. Monocytes were found to be enriched in OSA, and could be elevated by CPAP therapy.

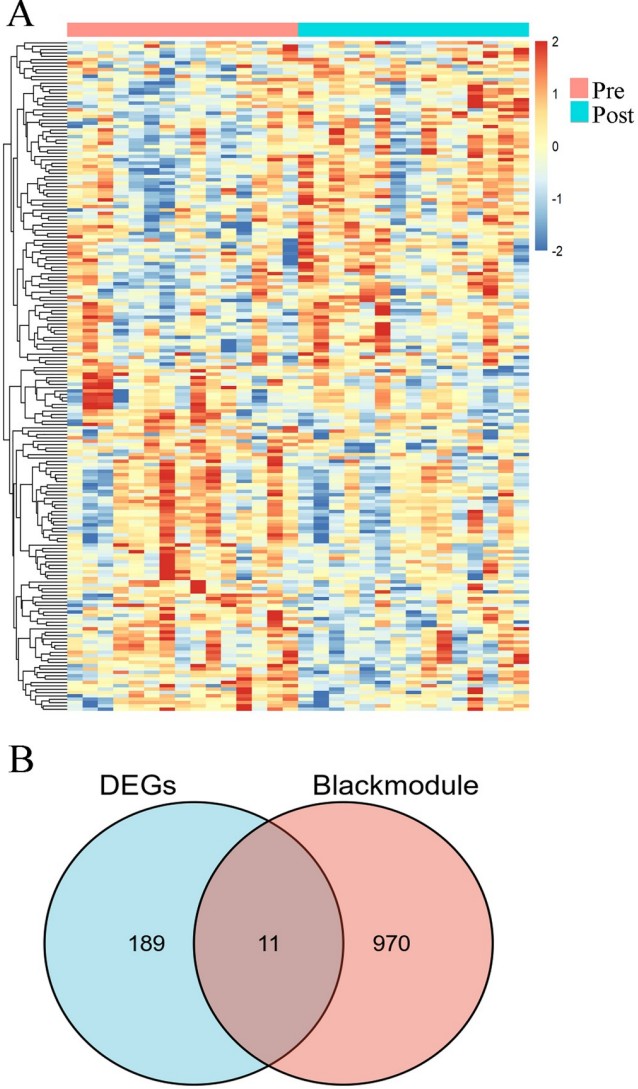

**Fig 3. Identification of hub genes by intersection of black module genes and top 200 DEGs.** (A) Heat map of the top 200 DEGs. (B) Venn plot of common genes.

Part of the hub genes are involved in immune function obviously. Immunoglobulin Lambda Variable 1–40 (IGLV1-40) is a subtype of immunoglobulin light chains, and T cell receptor alpha variable 10 (TRAV10) is a subtype of T cell receptor (TR) alpha chain. V region of them participates in the antigen recognition [27]. CD22 is a receptor predominantly restricted to B cells and its main function is to inhibit BCR signaling [28]. However, direct proofs of closely relationship between the other genes and immunity are limited. For example, the ribosomal protein L10 (RPL10), also known as QM and DXS648, was firstly identified from human tumor cells as a tumor suppressor [29]. It belongs to a highly-conserved component of the large ribosomal subunit (60S) [30]. Odorant binding protein 2B (OBP2B) belongs to the lipocalin (LCN) family [31]. It played important roles in physiological processes by binding to and transporting small hydrophobic molecules including odorants, retinoids, steroid hormones, and lipids [32]. Meanwhile, there are no related reports whether these 11 hub genes can be used as therapeutic efficacy biomarkers of OSA.

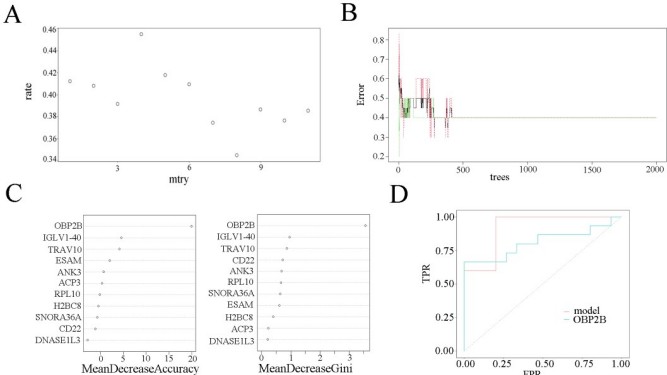

**Fig 4. Construction of random forest model.** (A) The scatter plot of the false-positive rate. The vertical axis represents the false positive rate, and the horizontal axis represents mtry index (from 1 to 11). (B) Relationship between the related errors and the number of decision trees. The vertical axis represents the related errors, and the horizontal axis represents the number of decision trees. (C) The scatter plot of the random forest variable based on the index of mean decrease accuracy (the left panel) and mean decrease Gini (the right panel), respectively. (D) Evaluation of the prediction efficiency of the random forest model via ROC curve. Red line represents the random forest model, and green line represents OBP2B gene.

It had been demonstrated that immune function played pivotal role in the development of OSA. Circulating monocytes [33] of OSA patients were activated and cytotoxic to endothelial cells. Specifically, monocytes in patients with OSA exhibited an immunosuppressive phenotype (high levels of HIF-1α, TGF-β, IL-10 and VEGF and decreased levels of IL-12p40), and CPAP treatment partially restored the impaired immune phenotype [34]. The results of CIBERSORT indicated that monocytes were enriched in OSA. Moreover, RPL10 and OBP2B had negative correlation with monocytes.

Nevertheless, the limitations of this study should also be clearly pointed out. Firstly, no direct experiments were performed to validate the prediction model of hub genes. Secondly, further proofs about the detailed molecular mechanisms are necessary. Finally, the small patient numbers and scant analytical methods may limit the predictive capability of the present model.

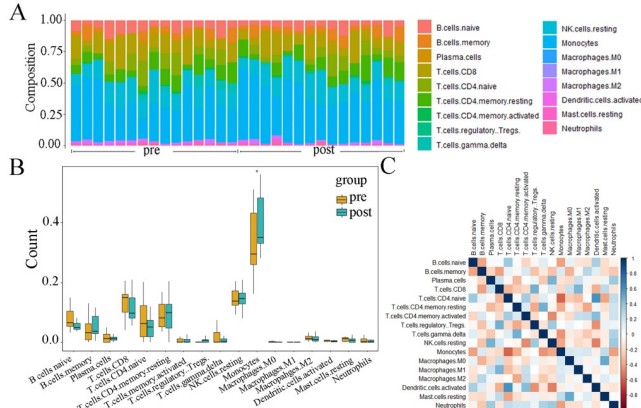

**Fig 5. Evaluation and visualization of immune cell infiltration.** (A) Stacked barplot diagram of immune cell types of 30 samples in GSE133601. (B) Boxplot diagram of immune cells. (C) Correlation heat map of immune cells. Blue represents a positive correlation, red represents a negative correlation. The darker the color, the stronger the correlation.

## Conclusion

We identified 11 hub genes via integrated bioinformatics methods and constructed a random forest model with them to predict therapeutic efficacy of CPAP for OSA. This paper also indicated monocytes may be related with the remission of OSA. The mechanism between immune cells and these key genes may be of great importance in the remission of OSA and related research of these genes may provide a new therapeutic target for OSA in the future.

## Author Contributions

**Conceptualization:** Yi Song.

**Data curation:** Cheng Fan.

**Formal analysis:** Cheng Fan, Yi Song.

**Funding acquisition:** Shiyuan Huang.

**Investigation:** Shiyuan Huang, Tianhui An.

**Methodology:** Yi Song.

**Project administration:** Cheng Fan, Yi Song.

**Resources:** Shiyuan Huang, Chunhua Xiang.

**Software:** Yi Song.

**Supervision:** Cheng Fan, Yi Song.

**Validation:** Shiyuan Huang, Chunhua Xiang.

**Visualization:** Cheng Fan, Yi Song.

**Writing – original draft:** Cheng Fan, Yi Song.

**Writing – review & editing:** Cheng Fan, Yi Song.

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
