## [Decision Letter · Decision Letter 0]

12 Aug 2021

PONE-D-21-23106

Identification of key genes and immune infiltration modulated by CPAP in Obstructive sleep apnea by integrated bioinformatics analysis

PLOS ONE

Dear Dr. Song,

Thank you for submitting your manuscript to PLOS ONE. After careful consideration, we invite you to submit a revised version of the manuscript that addresses the points raised during the review process.

We look forward to receiving your revised manuscript.

Kind regards,

Giannicola Iannella

Academic Editor

PLOS ONE

Journal Requirements:

4.Please include your full ethics statement in the ‘Methods’ section of your manuscript file. In your statement, please include the full name of the IRB or ethics committee who approved or waived your study, as well as whether or not you obtained informed written or verbal consent. If consent was waived for your study, please include this information in your statement as well. 

Reviewers' comments:

Reviewer's Responses to Questions

**Comments to the Author**

1. Is the manuscript technically sound, and do the data support the conclusions?

Reviewer #1: Yes

Reviewer #2: Yes

2. Has the statistical analysis been performed appropriately and rigorously? 

Reviewer #1: Yes

Reviewer #2: Yes

3. Have the authors made all data underlying the findings in their manuscript fully available?

Reviewer #1: Yes

Reviewer #2: Yes

4. Is the manuscript presented in an intelligible fashion and written in standard English?

Reviewer #1: No

Reviewer #2: No

5. Review Comments to the Author

Reviewer #1: Abstract

line 2, the sentencens are separated by a point. Please unify.

Background

- page 2, line 1, same of abstract line 2. Please unify the sentences.

- after line 1 add: The direct consequence of the intermittent hypoxia is an oxidative imbalance, with reactive oxygen species production, inflammatory cytokines (IL2, IL4, IL6), lipid peroxidation, and cell-free DNA production and please cite doi:10.3390/jcm10020277

- line 5, describe the findings of the study cited [5].

- line 7, add: however CPAP treatment could fail in patients with lingual tonsil hypertrophy and TORS could represent a valid therapeutic option for retrolingual airway collapse. please cite doi:10.1002/rcs.2106

- line ''Furthermore, immune cells infiltration was evaluated in OSA patients via CIBERSORT, and correlations between key genes and immune cells infiltration were calculated. '' Introduce first why you calculated immune infiltration and if there are studies that correlate it mention them in the text.

Materials and methods

describe the study design, year, protocol and cite a flow diagram to explain all the protocol with selection criteria.

Results

interesting and well written

Discussion

Line 1, cite papers on CPAP role in OSAS treatment discussing the main findings: doi:10.1007/s11325-014-1097-3

doi:10.1007/s11325-016-1419-8

- [11] please check the font

- '''Others have not been reported to have closely relationship with immunity.'' please cite the papers that states what you affirmed

Reviewer #2: The authors interestingly analyzed OSAS related genes correlating the outcomes with CPAP treatment. The paper has scientific soundness; however, minor corrections are required.

- English editing is necessary for both grammatic and punctuation (.And, patten, time‐ consuming);

Abstract

- before the sentence ''in conclusion.. report briefly the results found.

Background

- CPAP is not to date the only possible treatment in patients with obstructive sleep apnea. Different new technologies have been developed according to the profiles of the obstruction sites identified at Drug Induced Sleep Endoscopy, including different techniques from barbed wire pharyngoplasty up to robotic surgery in retrolingual collapses. Please cite doi: 10.1002 / rcs.2106 and doi: 10.1007 / s00405-020-05883-2

- please clarify the study protocol with a figure or a flowchart describing all the procedures performed;

Discussion

- cite references that support what stated in the sentence: ''The core genes of the clinically significant modules were supposed to be important in the remission of OSA''

the same with the sentence: '' Others have not been reported to have closely relationship with immunity. ...

- please read this review and modify the concept. In reality different biomarkers have been demonstrated in the literature: '' However, there are no related reports whether these genes can be used as therapeutic efficacy biomarkers of OSA. ''

- please read these metaanalysis and modify the concept. In reality different biomarkers have been demonstrated in the literature: '' However, there are no related reports whether these genes can be used as therapeutic efficacy biomarkers of OSA. ''

Imani MM, Sadeghi M, Khazaie H, Emami M, Sadeghi Bahmani D, Brand S. Evaluation of Serum and Plasma Interleukin-6 Levels in Obstructive Sleep Apnea Syndrome: A Meta-Analysis and Meta-Regression. Front Immunol. 2020 Jul 21; 11: 1343. doi: 10.3389 / fimmu.2020.01343. PMID: 32793188; PMCID: PMC7385225. and the paper

Rezaei F, Abbasi H, Sadeghi M, Imani MM. The effect of obstructive sleep apnea syndrome on serum S100B and NSE levels: a systematic review and meta-analysis of observational studies. BMC Pulm Med. 2020 Feb 5; 20 (1): 31. doi: 10.1186 / s12890-020-1063-8. PMID: 32024492; PMCID: PMC7003338.

- discuss at the end study's limitation

6. PLOS authors have the option to publish the peer review history of their article (what does this mean?). If published, this will include your full peer review and any attached files.

Reviewer #1: No

Reviewer #2: No

---

## [Author Response · Author response to Decision Letter 0]

22 Aug 2021

Review 1

Comment: (Abstract) line 2, the sentences are separated by a point. Please unify.

Respond: Correction has been completed.

Comment: (Background) page 2, line 1, same of abstract line 2. Please unify the sentences. after line 1 add: The direct consequence of the intermittent hypoxia is an oxidative imbalance, with reactive oxygen species production, inflammatory cytokines (IL2, IL4, IL6), lipid peroxidation, and cell-free DNA production and please cite doi:10.3390/jcm10020277

Respond: Correction has been completed.

Comment: line 5, describe the findings of the study cited [5].

Respond: The study cited [5] is a review conducted by Klaudia et al. We have cited the results from the review to prove our opinion “CPAP may exert various additional therapeutic benefits beyond its improvement of the oxygen supply”. 

Comment: line 7, add: however CPAP treatment could fail in patients with lingual tonsil hypertrophy and TORS could represent a valid therapeutic option for retrolingual airway collapse. please cite doi:10.1002/rcs.2106

Respond: Correction has been completed.

Comment: line ''Furthermore, immune cells infiltration was evaluated in OSA patients via CIBERSORT, and correlations between key genes and immune cells infiltration were calculated. '' Introduce first why you calculated immune infiltration and if there are studies that correlate it mention them in the text.

Respond: The reason why we calculated immune infiltration is that CIBERSORT is a widely used analysis tool using microarray data or RNA-seq data to investigate the expression profile of 22 types of immune cells and to calculate the proportions of each type of immune cells in the samples. We have already added the reason and cited related paper in the text.

Comment: (Materials and methods) describe the study design, year, protocol and cite a flow diagram to explain all the protocol with selection criteria.

Respond: Correction has been completed. A flow chart has been added as Fig1.

Comment: (Results) interesting and well written

Respond: Thanks for the comment.

Comment: (Discussion) Line 1, cite papers on CPAP role in OSAS treatment discussing the main findings: doi:10.1007/s11325-014-1097-3

doi:10.1007/s11325-016-1419-8

Respond: CPAP is one of the most effective therapies used in OSA. We have already cited related papers to prove it in revised manuscript. 

Comment: [11] please check the font

Respond: Correction has been completed.

Comment: '''Others have not been reported to have closely relationship with immunity.'' please cite the papers that states what you affirmed

Respond: To be more accurate, direct proofs of closely relationship between the other genes and immunity are limited, compared with IGLV1-40, TRAV10 and TRAV10. Among 11 hub genes, IGLV1-40, TRAV10 and TRAV10 encode proteins which are part of immunoglobulin, T cell or B cell, so they have closely relationship with immunity obviously. However, proteins encoded by other 8 genes do not have direct relationship with immunity. At least, they are not part of the key proteins or cells involved in classic immunity pathway and reports about their roles in immunity are limited.

Reviewer 2 

Comment: English editing is necessary for both grammatic and punctuation (.And, patten, time‐ consuming);

Respond: Corrections have been completed.

Comment: (Abstract) before the sentence ''in conclusion.. report briefly the results found.

Respond: The result is illustrated ''Eleven hub genes (TRAV10, SNORA36A, RPL10, OBP2B, IGLV1-40, H2BC8, ESAM, DNASE1L3, CD22, ANK3, ACP3) were traced and used to construct a random forest model to predict therapeutic efficacy of CPAP in OSA with a good performance with AUC of 0.92. Monocytes were found to be related with the remission of OSA and partially correlated with the hub genes identified''.

Comment: (Background) CPAP is not to date the only possible treatment in patients with obstructive sleep apnea. Different new technologies have been developed according to the profiles of the obstruction sites identified at Drug Induced Sleep Endoscopy, including different techniques from barbed wire pharyngoplasty up to robotic surgery in retrolingual collapses. Please cite doi: 10.1002 / rcs.2106 and doi: 10.1007 / s00405-020-05883-2

Respond: We have added related sentences at the end of the first paragraph in background. CPAP is not to date the only possible treatment in patients with obstructive sleep apnea. Other treatments (e.g. Surgery) could also be an option.

Comment: please clarify the study protocol with a figure or a flowchart describing all the procedures performed;

Respond: We have added a flow chart (Fig1) to describe the procedure of the study.

Comment: (Discussion) cite references that support what stated in the sentence: ''The core genes of the clinically significant modules were supposed to be important in the remission of OSA''. the same with the sentence: '' Others have not been reported to have closely relationship with immunity. 

Respond: Sentences mentioned above are not expressed appropriately and we have corrected them. 

1. ''The genes in key module were correlated significantly with the clinical trait of CPAP treatment''

2. ''Direct proofs of closely relationship between the other genes and immunity are limited, compared with IGLV1-40, TRAV10 and TRAV10''. Among 11 hub genes, IGLV1-40, TRAV10 and TRAV10 encode proteins which are part of immunoglobulin, T cell or B cell, so they have closely relationship with immunity obviously. However, proteins encoded by other 8 genes do not have direct relationship with immunity. At least, they are not part of the key proteins or cells involved in classic immunity pathway and reports about their roles in immunity are limited.

Comment: please read this review and modify the concept. In reality different biomarkers have been demonstrated in the literature: '' However, there are no related reports whether these genes can be used as therapeutic efficacy biomarkers of OSA. '' 

Imani MM, Sadeghi M, Khazaie H, Emami M, Sadeghi Bahmani D, Brand S. Evaluation of Serum and Plasma Interleukin-6 Levels in Obstructive Sleep Apnea Syndrome: A Meta-Analysis and Meta-Regression. Front Immunol. 2020 Jul 21; 11: 1343. doi: 10.3389 / fimmu.2020.01343. PMID: 32793188; PMCID: PMC7385225. and the paper

Rezaei F, Abbasi H, Sadeghi M, Imani MM. The effect of obstructive sleep apnea syndrome on serum S100B and NSE levels: a systematic review and meta-analysis of observational studies. BMC Pulm Med. 2020 Feb 5; 20 (1): 31. doi: 10.1186 / s12890-020-1063-8. PMID: 32024492; PMCID: PMC7003338.

Respond: To be more accurate, there are no related reports whether these 11 hub genes can be used as therapeutic efficacy biomarkers of OSA. From papers cited by reviewer, we knew that IL-6, S100B and NSE were reported to be biomarker of OSA. However, we did not find related papers whether 11 hub genes (TRAV10, SNORA36A, RPL10, OBP2B, IGLV1-40, H2BC8, ESAM, DNASE1L3, CD22, ANK3, ACP3) can be used as therapeutic efficacy biomarkers of OSA

Comment: discuss at the end study's limitation

Respond: study's limitation was discussed at the end of discussion part.

''Nevertheless, the limitations of this study should also be clearly pointed out. Firstly, no direct experiments were performed to validate the prediction model of hub genes. Secondly, further proofs about the detailed molecular mechanisms are necessary. Finally, the small patient numbers and scant analytical methods may limit the predictive capability of the present model.''

---

## [Editor Report · Decision Letter 1]

1 Sep 2021

Identification of key genes and immune infiltration modulated by CPAP in Obstructive sleep apnea by integrated bioinformatics analysis

PONE-D-21-23106R1

Dear Dr. Song,

We’re pleased to inform you that your manuscript has been judged scientifically suitable for publication and will be formally accepted for publication once it meets all outstanding technical requirements.

Kind regards,

Academic Editor

PLOS ONE

Additional Editor Comments (optional):

very interesting study.

The authors well replay to the comments of the reviewers. I believe that this paper is suitable of publication on PLOS ONE.

Best regards
---

## [Editor Report · Acceptance letter]

8 Sep 2021

PONE-D-21-23106R1 

Identification of key genes and immune infiltration modulated by CPAP in Obstructive sleep apnea by integrated bioinformatics analysis 

Dear Dr. Song:

I'm pleased to inform you that your manuscript has been deemed suitable for publication in PLOS ONE. Congratulations! Your manuscript is now with our production department. 

Kind regards, 

on behalf of

Dr. Giannicola Iannella 

Academic Editor

PLOS ONE